# Computer Simulation of a Surface Charge Nanobiosensor with Internal Signal Integration

**DOI:** 10.3390/bios11100397

**Published:** 2021-10-16

**Authors:** Dmitry Dyubo, Oleg Yu. Tsybin

**Affiliations:** Institute of Electronics and Telecommunications, Peter the Great St. Petersburg Polytechnical University, Polytechnicheskaya, 29, 195251 St. Petersburg, Russia; otsybin@rphf.spbstu.ru

**Keywords:** surface charge, nano, biosensor, computer simulation, COMSOL Multiphysics

## Abstract

The ionized states of molecular analytes located on solid surfaces require profound investigation and better understanding for applications in the basic sciences in general, and in the design of nanobiosensors, in particular. Such ionized states are induced by the interactions of molecules between them in the analyzed substance and with the target surface. Here, computer simulations using COMSOL Multiphysics software show the effect of surface charge density and distribution on the output generation in a dynamic PIN diode with gate control. This device, having built-in potential barriers, has a unique internal integration of output signal generation. The identified interactions showed the possibility of a new design for implementing a nanobiosensor based on a dynamic PIN diode in a mode with surface charge control.

## 1. Introduction

Methods and technology for the sensitive detection, characterization, and monitoring of biomolecules (proteins, lipids, DNA, and RNA molecules) and other biochemical and biophysical analytes are urgently needed in the life sciences, clinical diagnostics, the pharmaceutical sciences, the engineering of substances and materials, environmental research, and food control [1,2,3,4,5,6].

Nanobiosensors are analytical devices that combine a biologically sensitive element with a nanostructured transducer, and are widely used for molecular detection. They show certain advantages [7,8,9,10,11,12,13] due to their inherent specificity, simplicity, and quick responses. By design, nanobiosensors are hybrid devices in which an organic object is embedded in a nano or microelectronic semiconductor device.

Although promising for biomimetic materials applications, the binding of polypeptides to inorganic material surfaces and the mechanism of their interaction have been challenging to characterize. A few papers reported sequence–activity relationships of peptides interfacing with semiconductors and presented methodologies broadly applicable to the study of peptide–solid surface and molecular-molecular interactions [14,15].

The charge transport properties of proteins and other biomolecules provide opportunities to design and build the next generation of hybrid bioelectronic interfaces towards more efficient and biocompatible electronic devices. Most biomolecules have internal charges [16,17]. A semiconductor device based on the field effect makes it possible to detect variations in the charges of biological processes in real-time and in a non-invasive way. In such devices, one positive or one negative charge of a molecule electrostatically interacts with one electron charge in a semiconductor device. Thus, the induced change in the semiconductor’s surface potential acts as the gate voltage in a traditional field-effect transistor. In this case, there is a change in the current in the channel between the source and the drain. This current can be measured, and thus detect the presence of a bound analyte. The advantages of a field-effect transistor as a single device for surface charge detection include its small size and practical design. However, when using them as electrical sensors, there are problems associated with low signal intensity, signal-to-noise ratio, and relatively low sensitivity.

Surface potential and surface-charge-sensing biomolecular techniques have attracted a wide range of research interest in recent decades [18,19,20,21]. The ionized states of molecular analytes on solid surfaces are neither well-studied nor understood. Therefore, these ion–surface interactions require deeper investigation for basic science applications and the development of nanobiosensors [22,23,24,25,26,27].

The underlying cause for this charged layer creation on surfaces may be the separation of charges in the film due to the differences in the charge carriers’ mobility [28,29]. Protons have an abnormally high mobility and can produce H_3_O^+^ ions and their stable complexes [30,31]. In addition, such ions could be formed on a surface during the self-ionization reaction of water [28,29,30].

Another active ion is the NO^+^ cation, capable of creating strong bonds with water molecules and form stable clusters of NO^+^ + (H_2_O)_n_ [32].

The connection of water molecules with these surfaces and the formation of their corresponding ions in the film can also be interpreted in terms of a simple acid–base interaction, according to Lewis [33]: the water molecule acts as an electron donor (Lewis base), and the substrate as an electron acceptor (Lewis acid).

Similar phenomena are known to occur in semiconductor biosensors [9].

The actual technical task is to design more effective, hybrid, nanoelectronic, and charge-sensitive devices, which operate with a label-free biomolecular analyte bound to the target surface.

The recently presented PIN photodiode possesses a unique internal integration function of the time-dependent signal, thus providing high sensitivity, low noise, and a wide dynamic range [34]. The diode contains an embedded potential barrier formed by a gate-controlled depletion region. In a dynamical anode voltage regime, an output analog signal is proportional to the energy dose of an incident electromagnetic wave (visible light). This phenomenon includes photo- and thermo-generation of electron-hole pairs in the semiconductor bulk, and gate control of potential barrier magnitude, which, in turn, could regulate the charge carrier flow.

In our previous PIN diode study, both the temperature dependence and the effect of the presence of external particles on the substrate surface were revealed [35]. When a molecular film appears on the surface between the gate and the cathode, bound charges appear on that surface as well, which create the corresponding electric fields.

Here, we studied the mechanism of the influence of surface charge on signal generation in a PIN diode containing embedded potential barriers. The appropriate mechanism of the barriers’ height modulation was revealed and estimated by means of computer simulation software, COMSOL Multiphysics. The obtained results show some principal possibilities of using this device as a highly sensitive semiconductor nanobiosensor which performs signal integration, controlled by surface charge.

## 2. Methods

The initial state of the dynamical PIN diode was when the anode and the gate were kept under negative bias voltage [34]. The measuring cycle started when the anode negative bias was switched to the forward one (Figure 1). The following figure represents the operating scheme where only one potential barrier is shown.

Initially, electrons and holes, trapped by electrostatic potential barriers above Fermi level, move faster near the depletion regions’ borders. The charge carriers outside of the depletion region are relatively slow due to the absence of a strong static electric field. During the filling of depletion regions by current flowing from the cathode, potential barriers become lower and narrower due to charge carriers accumulating under the gate and shielding the electric field. The device switches on from a low zero-field current to a high-current state after a certain triggering delay. Switching occurs abruptly due to tunnel breakdown through the narrow residual barrier. The positive feedback effect of electrons and holes mutually enhances motion between anode and cathode, which intensifies the described process.

Without external irradiation, the zero-field thermo-generated current *I*_thermo_ can determine the value of self-triggering time,
(1)$$T_{\mathrm{trig}}=\frac{Q_{c}}{I_{\mathrm{thermo}}},$$
where $Q_{c}$ is a critical value of the accumulated charge. Further, the device switches on abruptly to a high-current state after a certain time delay.

Since the irradiating intensity decreases the triggering time, one can obtain a higher measurement sensitivity by changing the external irradiating intensity. Under light illumination in a visible wavelength range of approximately 400 to 800 nm, photo-generated electron-hole pairs are separated by an internal electric field inside a semiconductor target. Electrons drift under the gate, whereas holes move to the cathode. Over time, the accumulation of electrons under the gate starts to shield the electric field. Finally, the decrease in the depletion region de-isolates the anode, which emits holes that can lower the cathode barrier by locally increasing the potential.

The total critical charge $Q_{c}\left(\tau \right)$ value, accumulated by the flow of summed thermo- generation and photo-generation currents, is described by the following expression [37]:(2)$$Q_{c}\left(\tau \right)\sim \int_{0}^{\tau }dt\cdot T^{2}exp\left(\frac{-U_{b}-\Delta U_{b}\left(\Phi ,T,t\right)}{kT}\right),$$
where the employed parameters are: time *t*, pulse duration τ, temperature *T*, Boltzmann constant *k*, and embedded potential barrier height *U*_b_. Potential shift Δ*U*_b_ (*Φ*,*T*,*t*) describes an instantaneous effective decrease in the barrier height due to space charge neutralization by thermo- and photo-generated carriers, where *Φ* is the visible light external irradiation dose.

The model corresponding to Formula (2) considers the contribution of the photocurrent to barrier charge neutralization, modulation of the barrier height by photogenerated carrier charge, and the temperature dependence of the Boltzmann barrier height [36]. Using Expression (2), one can estimate the dependence of the barrier height on the charge accumulation leading to the internal signal generation.

The Semiconductor Module of COMSOL Multiphysics solves Poisson’s equation in conjunction with the continuity equations for charge carriers. The mobility model defines both electron and hole mobilities. Constant and user-defined doping profiles can be specified, or an approximate Gaussian doping profile can be used. In the model employed here, the user-defined profile was set.

A stationary study type was applied, where the variables used were the electric potential V, electron N, and the hole P concentration. Dealing with this set of variables was sufficient for this study’s task, where the potential and the electric field as well as the electron and hole concentration distributions were studied. Figure 2a–d show the calculated COMSOL 2D distribution of potential inside the model device, with different charges immobilized on the surface between gate and cathode.

Evidently, even at a low surface charge of approximately 0.01 in one monolayer (the thickness of the film in the picture is conditional, in fact, the film is infinitely thin), this is reflected in the results as a disturbed potential. For the shown device model, we realized COMSOL [38] computer modeling of potential barriers’ characteristics, controlled by a charged film situated in the area between cathode and anode on an Si or GaAs surface.

## 3. Results

We considered the dynamics of the processes occurring in the absence of external irradiation, which would lead to the generation of photoelectrons. Accordingly, the existence of two currents is possible, namely, the current caused by the thermal generation of electrons $I_{\mathrm{thermo}}$, and the leakage current $I_{\mathrm{leak}}$, caused by the action of the electric field at the cathode. The variations in the charge accumulated under the gate and the reduction in the depletion region are shown in the graphs below.

The potential shift ∆*U*(*Q*_s_) between the cathode potential and the minimum potential was considered. One can introduce the parameter $p=\left(\Delta U-\Delta U_{0}\right)/Q_{s}$, where $\Delta U_{0}$ ≈ 0.58 V for surface charge $Q_{s}=0$. This parameter remains almost constant for $Q_{s}$ when the surface charge is in the range from −10^−4^ to +10^−4^ C∙m^−2^ but decreases for absolute values more than 10^−4^.

The quasi-parabolic potential trap (Figure 3) controlled the injection of electrons from the cathode into the embedded depletion region. Further, one can determine the leakage current of electrons coming from the cathode n+ region into the potential well,
(3)$$I_{\mathrm{leak}}=I_{0}\cdot exp\left(\frac{\Delta U\left(Q_{s}\right)-\Delta U_{0}}{kT}\right)$$

The leakage current is locked by the potential shift Δ*U* < 0 and could be shifted by surface charges.

In the absence of both external illumination and thermo-generation, the switching time *T*_trig_ may be determined by the leakage current of electrons from the cathode area:(4)$$T_{\mathrm{trig}}=\frac{Q_{c}}{I_{\mathrm{leak}}}=\frac{Q_{c}}{I_{0}}\Delta exp\left(-\frac{pQ_{s}}{kT}\right)$$

In accordance with the analytical expression (4), Figure 4 demonstrates the linear dependences of the values $ln\left(T_{\mathrm{trig}}\right)$ on the surface charge density.

The calculated graphs with the potential shift value $\Delta U=p\cdot Q_{s}+\Delta U_{0}$ (Table 1) are shown on a logarithmic scale in Figure 4. Their linearity in a wide range of the surface charge density, as well as for different concentrations of ligands in the range of up to four orders, is in accordance with the approximation by the exponential function, and with expressions (2) and (4). The advantages of the weak doping of semiconductors at a level of 10^12^ cm^−3^ are obvious for obtaining the highest sensitivity of surface charge detection.

Further, it was calculated by means of the potential approximation function that the electric field has a steep dependence on the surface charge density for absolute minimum values of less than 5 × 10^−4^ C∙m^−2^, which determined the high sensitivity of the device as a sensor. Such dependencies were obtained at different levels of doping in a wide range (Figure 5).

The electric −5 × 10^−^^14^ field near the surface reached high values in the order of 10^4^ V∙cm^−1^ (see Figure 5). Such a field acts not only perpendicular to the surface but also along it. In a strong electric field, an ensemble of charged particles can be separated spontaneously along the surface into separate islands. Apart from the transverse field, a strong lateral electric field appears between the islands in the longitudinal direction along the surface.

The effect of a dashed, surface-charged film with (i) three and (ii) fourteen islands, into which the continuous homogeneous film was separated, is shown as an example (Figure 6, Figure 7 and Figure 8). Due to the action of an electric field transverse to the surface, the distribution of the electric potential in the volume of the semiconductor at different depth values demonstrates, in a certain way, the effect of the film island structure (Figure 6 and Figure 7). The type of the curve depends on the Y-position of the cutline along the substrate depth.

One can observe the expected gradual reduction in the surface charge effect along the substrate depth (Figure 7).

Figure 8 shows the electric field distribution in the target volume generated by the surface charge, with a different number of charged islands. The Coulomb splitting of the islands was observed not only for low numbers (Figure 8a), but even for high numbers (Figure 8b). The presence of a high peak at *X* = 1.2 μm (Figure 8) can be explained by an edge effect at the contact point between the metal gate and substrate.

## 4. Discussion

For the considered device, whose operation scheme is shown in Figure 1, the dependence of potential barriers inside the semiconductor volume, Δ*U* = *p* · *Q*_s_, *p* ≈ *const* on surface charge, $Q_{s}$, was revealed. Thus, the relationship between the measured signal, in the form of the current pulse duration (triggering time), and the surface charge value was determined in expression (4) and the graphs in Figure 4. Employing these results, one can measure the switching time $T_{\mathrm{trig}}$ for calculation of the instantaneous surface charge density $Q_{s}$ and of its dynamics at a frequency range lower than $1/T_{\mathrm{trig}}$.

The electric field near the surface reached high values in the order of 10^4^ $V\cdot {\mathrm{cm}}^{-1}$, which can be estimated by the same method by which the dependencies are shown in Figure 5. In such high electric fields, generated by the absence of neutralized charge in the surface molecular films, appropriate biophysical and biochemical reactions could be activated. Even though the details of this phenomenon have not been determined, the very fact that molecular charging on the surface occurs, especially in the presence of water molecules, corresponds to modern ideas about the adsorbed layer. Surface charges generate an electric field that changes the appearance of the potential barriers in semiconductor targets. Comparing Figure 6, Figure 7 and Figure 8, it can be seen that the separation of surface charge into islands does not violate the concept of the influence of this charge on the structure of potential barriers in the volume of the substrate, and, accordingly, on the emission of current from the cathode.

## 5. Conclusions

The identified interactions suggest a new possible design of a nanobiosensor based on the dynamic PIN diode. This new device is a hybrid one, in which the transducer is a microelectronic, semiconducting diode with an additional gate electrode, operating in a dynamic mode. This device has a built-in molecular, including biomolecular, object, in the form of a film on the semiconductor surface in the area surrounding the gate.

The hybrid device made it possible to detect and measure bound surface charges, and monitor ionization of molecules, which were either self-generated or induced by external influences in the film. High sensitivity and low noise levels were achieved through the ability of the device to provide the internal summation, and integration of the transferred space charges that enter the electrical potential well. The internal integration distinguishes this device from known nanobiosensors based on transistors and other diodes. As a result, the integral dose with a background of the averaged noise as well as the enhanced signal-to-noise ratio were measured instead of the instantaneous signal with the background of the real noise. On the basis of the measurements, specific surface analytes and their reactions can be determined, and knowledge of the ionization characteristics of molecules, including biomolecules, can be developed.

## Figures and Tables

**Figure 1 biosensors-11-00397-f001:**
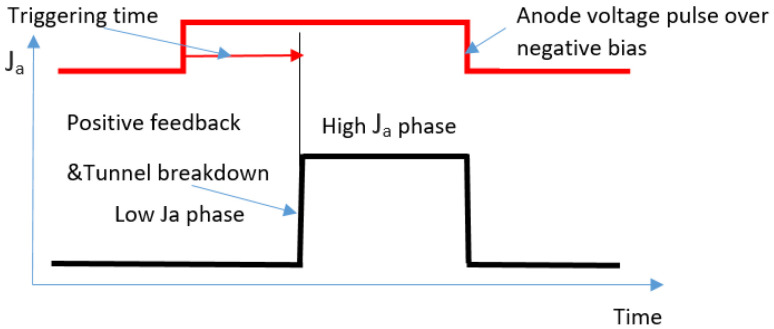
Operating scheme of dynamical semiconductor sensor. *J_a_* is the anode current. Reprinted with permission from ref. [36]. Copyright 2019 IEEE.

**Figure 2 biosensors-11-00397-f002:**
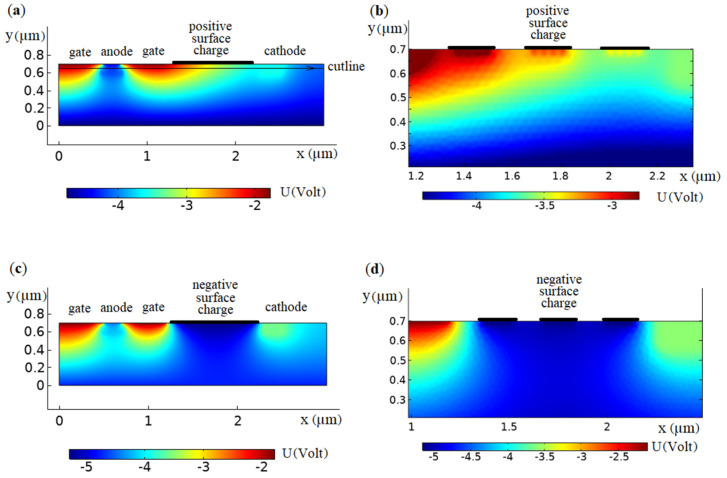
Potential distribution inside the *p*-type GaAs target (concentration of dopant is *p* = 10^12^ cm^−3^) with positive surface charge located between cathode and anode of the model device with uniform surface density *ρ*, $C\cdot m^{-2}$ (**a**,**c**) and three surface islands (**b**,**d**): (**a**) +10^−3^; (**b**) +10^−2^; (**c**) −10^−3^; (**d**) −10^−2^. Thin oxide or dielectric layer was located between the surface charge and the bulk. Cathode voltage *U*_ct_ = 0.5 V, anode voltage *U*_an_ = 1 V, gate voltage *U*_g_ = 3 V. Color gradation presents the potential distribution range, as shown in the color lines.

**Figure 3 biosensors-11-00397-f003:**
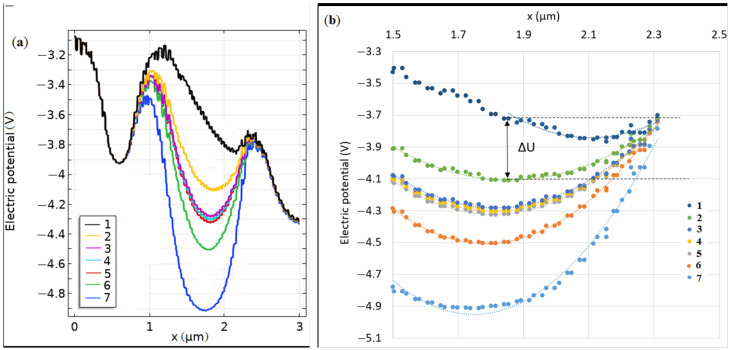
Electric potential distribution at different surface charge density values on two scales on the *X*-axis. The cutline *Y* = 0.5 μm, *U*_ct_ = 0.5 V, *U*_an_ = 1 V, *U*_g_ = 3 V. The target is GaAs, *p*-type, the dopant concentration equal *p* = 10^12^ cm^−3^. Barrier −height Δ*U* was determined for each curve between the cathode potential and the minimum potential (shown for curve number 2). Surface charge values are shown in Table 1 below.

**Figure 4 biosensors-11-00397-f004:**
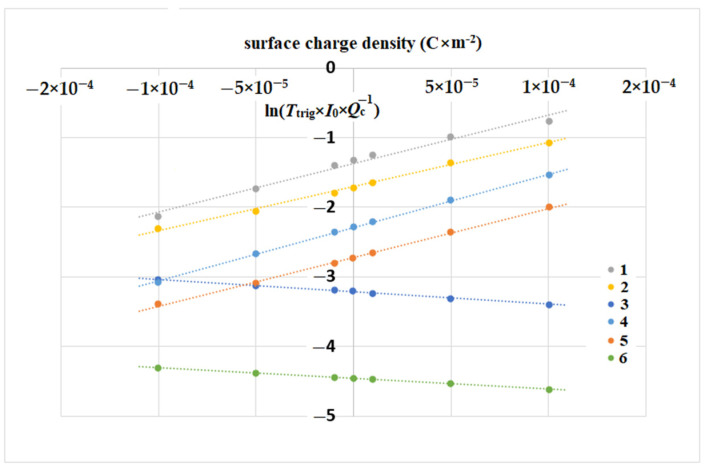
The dependence of normalized triggering time on the surface charge density $Q_{s}$, target Si and GaAs, *p*-type. *U*_ct_ = 0.5 V, *U*_an_ = 1 V, *U*_g_ = 3 V. The dopant concentration, [cm^−3^], equal for p-Si: (**1**) 10^12^; (**2**) 10^14^; (**3**) 10^16^; and for p-GaAs: (**4**) 10^12^; (**5**) 10^14^; (**6**) 10^16^.

**Figure 5 biosensors-11-00397-f005:**
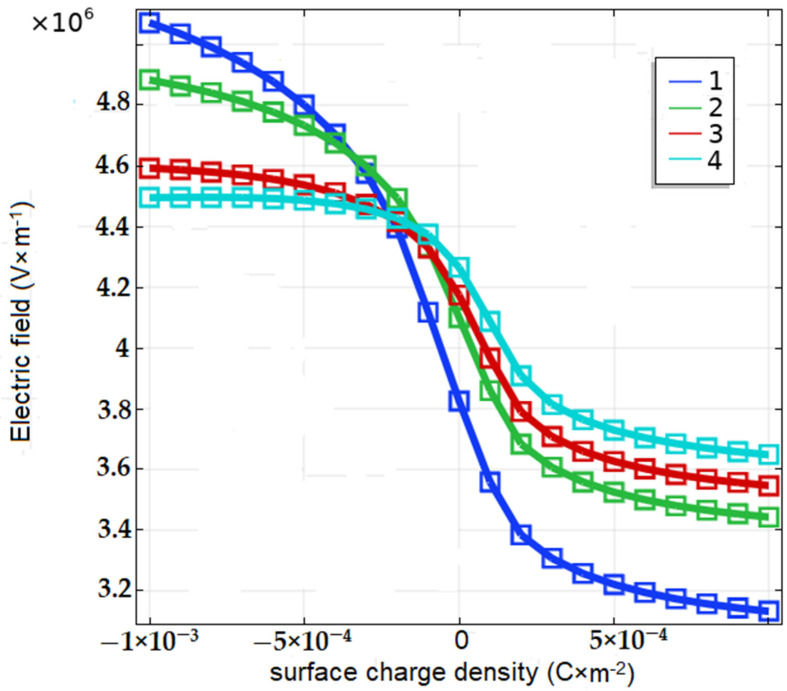
The dependence of electric field on the surface charge density ρ at different levels of p-dopant. Cutpoint *X* = 2.3 μm, *Y* = 0.5 μm. GaAs, *p*-type, cm^−3^: (**1**) 10^12^, (**2**) 10^14^, (**3**) 5 × 10^14^, (**4**) 10^15^.

**Figure 6 biosensors-11-00397-f006:**
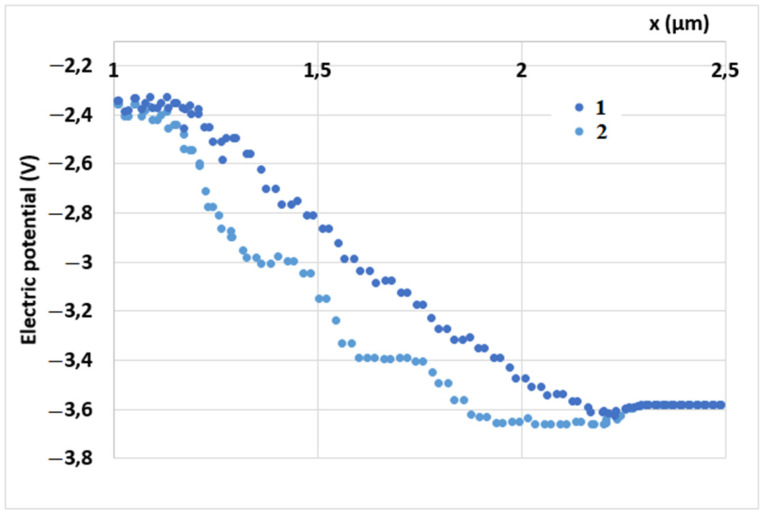
Electric potential distribution. Different number of surface charge $Q_{s}$ segments. *U*_ct_ = 0.5 V, *U*_an_ = 1 V, *U*_g_ = 3 V. Cutline *Y* = 0.65 μm, GaAs, *p*-type, *p* = 10^12^ cm^−3^. Surface charge density *ρ* = 10^−3^ $C\cdot m^{-2}$. Uniform charged film (**1**) was divided for three islands (**2**), as shown in Figure 2 above.

**Figure 7 biosensors-11-00397-f007:**
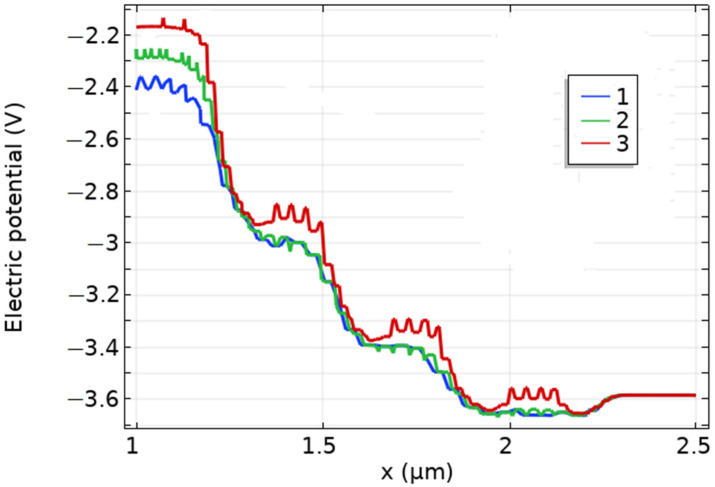
Electric potential distribution. Different cutlines: (**1**) *Y* = 0.65 μm, (**2**) *Y* = 0.665 μm, (**3**) *Y* = 0.685 μm (located closely to the surface *Y* = 0.700 μm). GaAs, *p*-type, *p* = 10^12^ cm^−3^. Three surface charge segments = 3. *U*_ct_ = 0.5 V, *U*_an_ = 1 V, *U*_g_ = 3 V. Surface charge density *ρ* = 10^−3^ $C\cdot m^{-2}$.

**Figure 8 biosensors-11-00397-f008:**
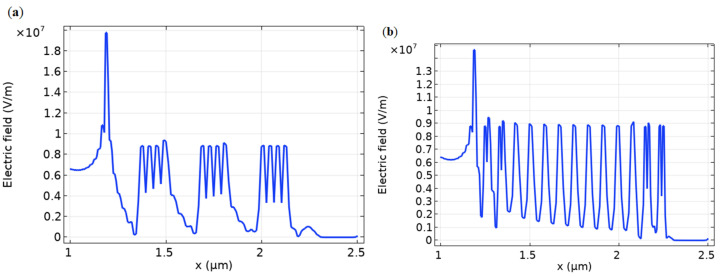
Electric field with charged islands on the surface. The number of charged islands is 3 pieces (**a**), and 14 pieces (**b**). *U*_ct_ = 0.5 V, *U*_an_ = 1 V, *U*_g_ = 3 V. Cutline *Y* = 0.70 μm. GaAs, *p*-type, *p* = 10^12^ cm^−3^. Surface charge density *ρ* = 10^−3^ $C\cdot m^{-2}$ is situated at each island.

**Table 1 biosensors-11-00397-t001:** Surface charge values and corresponding approximation functions for the potential shift.

	$\mathrm{Surface}\;\mathrm{Charge}\;Q_{s},\;C\cdot m^{-2}$	Approximation Function	The Potential Difference ΔU, V	$p=\left(\Delta U-\Delta U_{0}\right)/Q_{s}$
1	10^−3^	2.3987*x*^2^ − 2.9071*x* − 2.9603	0.163415	591.75
2	10^−4^	1.6546*x*^2^ − 1.1781*x* − 3.8982	0.390072	1977.19
3	10^−5^	2.1176*x*^2^ − 1.3315*x* − 4.0729	0.558116	1964.37
4	0	2.1703*x*^2^ − 1.3498*x* − 4.0924	0.577734	0
5	−10^−5^	2.2232*x*^2^ − 1.3683*x* − 4.1118	0.597378	1961.84
6	−10^−4^	2.7016*x*^2^ − 1.5384*x* − 4.2868	0.775454	1876.63
7	−10^−3^	3.4629*x*^2^ − 1.7135*x* − 4.7383	1.16948	414.32

## Data Availability

The data that support the findings of this study are available from the corresponding author upon reasonable request.

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
