# Peer review of "Computer Simulation of a Surface Charge Nanobiosensor with Internal Signal Integration"

_biosensors, 2021, doi:10.3390/bios11100397_

Round 1

Reviewer 1 Report

This paper presents a computer simulation of a surface charged nanobiosensor with a signal internal integration method. In general, this paper lacks enough scientific significance and innovation, which does not fit high-impacted Biosensors. In addition, verbiage and data analysis is beyond professional. In the light of these facts, I could not suggest acceptance as it is. 

I recommend the authors to ask a native speaker to edit the whole paper as there are a bunch of lexical and grammatical errors throughout the article, rendering unsuccessful information delivery to readers. 

There are some problems regarding cleanness of data visualization. Most images require further revision. Ja in Figure 1 needs explanation in the caption; 3, 4, and 5 are hard to distinguish in Figure 2. Also, please use legends instead of 1,2,3 to label the curves. Significant figures are not consistent, either. 

Author Response

The authors are grateful for the review and recommendations on editing the manuscript. We made an attempt to meet the requirements suggested by the reviewer. The correction from the lexical and grammatical point of view was also made. The legends instead of labels were used at the graphs. 

Reviewer 2 Report

This paper, prior to be accepted, needs necessary revision in language and data description.
In its current form, the article in places looks incomprehensible, since it looks like it has been translated from Russian scientific slang.

Other question which needs clarification:

1.Is it possible to clearly describe the capabilities of COMSOL for the tasks of this paper? What the given code can exactly take into account and the calculation will be appropriate and what is beyond its capabilities

2.How does the efficiency of electron-hole pair production and their mutual recombination is taken into account as a function of wavelength and intensity, as well as the depth of penetration of photons?

3. How do the results depend on the type of dopant and its charge state?

4. How do the results depend on surface defectiveness and surface termination?

5. The relevance of the work is usually confirmed by appropriate references.  However, the first 9 references  are all pretty old. Is it possible to further explain that the work is really relevant?

Author Response

The authors are grateful for the review and recommendations on editing the manuscript. The edits along the manuscript were done to meet the requirement of the reviewer. The correction from the lexical and grammatical point of view was also made.

  1. The description added to the text.
  2. In contrast to the usual application of the device as a photodiode, the sensor is based only on the relationship between the charge field and the barrier height, which determines the injection of carriers from the cathode, and, as a result, the signal pulse duration. Therefore, the consideration of the electron-hole pair production efficiency was not required. The use of additional lighting will be considered in the future. Thereby the dependence of the electron-hole pair production on wavelength and intensity, as well as depth of penetration of photons were not taken into account. For the recombination of thermogenerated electron-hole pairs the Shockley-Read-Hall model was used.
  3. One should consider the calculated graphs with the potential shift value ∆U shown on a logarithmic scale in Figure 4. The advantages of weak doping of semiconductors at a level of 1012 cm-3 are obvious for obtaining the highest sensitivity of the surface charge detection.
    The model considered p-doped substrate which approved the expected effect. According to estimates, it is possible to obtain a similar effect for the n-type substrate.
  1. Nanostructured surfaces are of great interest. Such surfaces are a subject for further study. Defects on the surface, as is known, lead to a concentration of charge on themselves, so the islands were considered in the model. According to the data obtained, we can expect that defects will increase the sensitivity.
  2. The authors made an ettempt to explain more clear the relevance of this work in the edited intoduction part.

Round 2

Reviewer 1 Report

The authors addressed all my questions so I suggest acceptance.